# Characterization of binding between model protein GA-Z and human serum albumin using asymmetrical flow field-flow fractionation and small angle X-ray scattering

Jaeyeong Choi[1], Marie Wahlgren[1], Vilhelm Ek[2], Ulla Elofsson[3], Jonas Fransson[2], Lars Nilsson[1], Ann Terry[4], Christopher A. G. Söderberg[3]*

1 Food Technology, Engineering and Nutrition, Faculty of Engineering LTH, Lund University, Lund, Sweden, 2 Swedish Orphan Biovitrum AB (Sobi), Stockholm, Sweden, 3 Division of Bioscience and Materials, RISE Research Institutes of Sweden, Lund, Sweden, 4 MAX IV Laboratory, Lund University, Lund, Sweden

* christopher.soderberg@ri.se

## Abstract

Protein-based drugs often require targeted drug delivery for optimal therapy. A successful strategy to increase the circulation time of the protein in the blood is to link the therapeutic protein with an albumin-binding domain. In this work, we characterized such a protein-based drug, GA-Z. Using asymmetrical flow field-flow fractionation coupled with multi-angle light scattering (AF4-MALS) we investigated the GA-Z monomer-dimer equilibrium as well as the molar binding ratio of GA-Z to HSA. Using small angle X-ray scattering, we studied the structure of GA-Z as well as the complex between GA-Z and HSA. The results show that GA-Z is predominantly dimeric in solution at pH 7 and that it binds to monomeric as well as dimeric HSA. Furthermore, GA-Z binds to HSA both as a monomer and a dimer, and thus, it can be expected to stay bound also upon dilution following injection in the blood stream. The results from SAXS and binding studies indicate that the GA-Z dimer is formed between two target domains (Z-domains). The results also indicate that the binding of GA-Z to HSA does not affect the ratio between HSA dimers and monomers, and that no higher order oligomers of the complex are seen other than those containing dimers of GA-Z and dimers of HSA.

## Introduction

For many biological drugs, a short half-life of the drug circulating in the blood is a problem. Primarily, this is due to metabolism or, for smaller proteins, kidney uptake. Short half-life will either require frequent dosing or the need for high doses and, in the worst case, can mean that the drug is not efficient enough to reach the market. All of these scenarios cause problems for patients, e.g. low patient compliance, increased need for high drug doses which leads to increased risks for adverse side effects, and finally a risk for low efficacy. The latter could also result in no viable treatment for a disease. Thus, strategies to increase blood circulation time for biological drugs are needed, and one such strategy is albumin binding.

**Data Availability Statement:** All relevant data are within the manuscript and its Supporting information files.

**Funding:** This project was funded by a Vinnova grant, supported by Swedish research council, awarded to UE and MW (grant number 2018-04730, www.vinnova.se). The funders had no role in study design, data collection and analysis, decision to publish, or preparation of the manuscript. The commercial company Swedish Orphan Biovitrum AB (Sobi) provided support in the form of salaries for authors VE and JF, but did not have any additional role in the study design, data collection and analysis, decision to publish, or preparation of the manuscript. The specific roles of these authors are articulated in the 'author contributions' section. Sobi also provided protein in the form of GA-Z to use for experiments.

**Competing interests:** The authors have declared that no competing interests exist. The commercial affiliation to Sobi through authors VE and JF does not alter our adherence to PLOS ONE policies on sharing data and materials.

Albumin is a key component of blood and, being the most abundant protein in blood, it is important for both osmolality and pH of the blood. Human serum albumin (HSA) is a compact globular protein with a molecular weight of 66.5 kDa. It consists of three homologous domains DI, DII and DIII, each containing an A and B subdomain [1] The pI of HSA will differ depending on whether the protein has bound fatty acids or not (pH 5.3 for fatty acid-free and around pH 4.7 when containing fatty acids) [2]. In solution, HSA often exists both as a monomer and dimer, with the monomer form being the dominant one [2, 3]. One of the key functions for HSA is to protect the body by binding otherwise toxic substances such as fatty acids, bilirubin, and nitric oxide to mention a few. The protein has several binding sites, for example for fatty acids [4, 5] as well as sites where peptide sequences can bind. Hence, there are several HSA binding proteins [6, 7].

Binding to HSA has been utilized for prolonging the recirculation time of various types of active substances [8]. Example of substances that bind to HSA are warfarin, diazepam and ibuprofen [2]. Traditionally, the hydrophobic binding sites (pockets) of HSA that bind fatty acids have been utilized. In the case of low molecular weight substances, the active pharmaceutical ingredient (API) itself has an affinity for these sites, or it has been covalently linked to a lipid tail. The latter has also been used for biologics. For example, insulin-like peptides have shown considerably increase in half-life when lipidated [9]. Ryberg *et al.* have shown that the tertiary structure of the HSA-lipidated peptide complex corresponds to hexamers, indicating that the structure of HSA can change as a consequence of API-binding [10].

Another strategy to bind proteins to Albumin is to attach an HSA binding domain to the peptide sequence, or domain, that binds to the therapeutic target. In 2010 Andersen *et al.* showed that a minimal albumin-binding domain (GA-domain) derived from streptococcal protein G could be used for binding to HSA and that the binding prolonged the half-life of this domain, as well as the fusion proteins of this domain with a so-called Affibody [11]. This has since been utilized successfully for several affibodies, as described in a recent review by Frejd and Kim [12]. For an overview of this type of HSA binding domains we refer to Nilvebrant and Hober [13] and for a more in-depth review of the use of HSA in drug delivery to Elsadek and Kratz [14] and Fanali *et al* [5]. In contrast to fatty acid binding, there is only one site on HSA that binds the GA-domain. This site is located in domain II of HSA [15], and in the literature we have found examples of equilibrium dissociation constants between 0.1 nM and 10 nM [11, 16].

The field of utilizing albumin binding domains for prolonged circulation of therapeutic proteins in blood is evolving quickly. Interestingly, Hein *et al. [17]* showed that there are changes to HSA structure in sites as far away as 24Å from the binding site of lidocaine. For this reason, we would like to rule out similar changes to HSA structure as a response to e.g. HSA dimerization or indeed interaction with HSA binding proteins. HSA dimerization could allosterically affect the interaction between HSA and HSA binding proteins, and similarly binding of HSA binding proteins could affect the monomer/dimer equilibrium of HSA. However, to our knowledge, no studies (to date) have described the dynamics of the tertiary structure of such complexes and how the monomer/dimer equilibrium of HSA affects the binding of a fusion protein containing one GA-binding domain and one targeting domain (Z-domain). Thus, the aim of this article is to demonstrate how such knowledge can be obtained using small angle X-ray scattering (SAXS) and asymmetric flow field-flow fractionation (AF4).

For this purpose, we used a model fusion protein (GA-Z) which primarily forms dimers in the formulation. The oligomeric state of the fusion protein may also affect the HSA binding and it can be expected that the monomeric form become more abundant upon dilution following injection, and therefore it is of importance also to establish in which form the protein

binds to HSA. The molecular weight of this protein is 11.5 kDa with an HSA binding domain of 5 kDa and a Z-domain of 6.5 kDa.

The crystal structure of HSA, as well as the complex between HSA [18] and the GA-binding domain [15] has been determined by others. The crystal structure of HSA indicates that the molecular dimensions of the protein are $80 \times 80 \times 30$ Å [18]. In the crystal structure between HSA and the GA-binding domain [14] the interface on HSA was found to be on the HSA DII domain. SAXS measurements do not give as detailed information as X-ray protein crystallography. On the other hand, they provide insight into the solution structure of the proteins, including information on the quaternary structure and, to some extent, the flexibility of the proteins and protein complexes. Furthermore, SAXS measurements in this study are performed in bulk solution, thus there is no need to produce protein crystals which may indeed at times be impossible especially for protein complexes. That being said, detailed analyses of SAXS data depends very much on previous work from other techniques. High resolution protein structures from NMR and X-ray crystallography can be used and combined in order to produce accurate models of e.g. protein complexes that on their own could not be isolated by previous techniques. However, as pointed out by Skou *et al.* [19] impurities or solutions containing both monomers and dimers will complicate the interpretation of the scattering profile. Therefore, it is important to first investigate the monodispersity of the sample before continuing with modeling of SAXS data. The use of a size-exclusion chromatography setup directly linked to the SAXS sample cell greatly aids the collection of monodisperse SAXS-data from more complex samples containing homo- or hetero-oligomers. The HSA structure at different pH was described by Olivieril and Craievich [20]. They interpreted the data as a monomer with a radius of gyration of 33.4 Å.

Size exclusion chromatography (SEC) is the most common method for the analysis of molecular weight (MW) and molecular weight distribution (MWD). Unfortunately, SEC has limitations when it comes to the exclusion limit or the permeation limit of column leading to underestimation or overestimation of MW. In addition, some molecules are known to undergo degradation by elution shear stress and/or be trapped in a stationary phase of SEC columns [1–3]. AF4 is a chromatography-like separation technique utilizing a channel void of stationary phase rather than a packed column. It is based on a combination of a longitudinal laminar flow of a carrier liquid through the separation channel and a perpendicular crossflow of the same carrier liquid. Due to the cross-flow, molecules and particles of different size will elute at different times and so the hydrodynamic radius of the sample can be determined from the elution profile. AF4 has several advantages over SEC. Due to the relatively gentle separation condition (e.g. low pressure and low shear stress), close to native condition, degradation of analytes is largely prevented during separation in AF4. [21, 22]. A detailed description of this method can be found elsewhere [23]. By combining AF4 with different types of detectors, for example multi-angle light scattering (MALS), differential refractive index (dRI) and UV, further information such as molecular weight, the radius of gyration and concentration in eluting peaks can be obtained.

## Materials and methods

The research data used to prepare this manuscript is available from the authors following a decision from the NextBioForm consortium, on a case to case basis.

### Materials

Sodium chloride (NaCl), disodium phosphate dihydrate (Na$_2$HPO$_4$·2H$_2$O), sodium dihydrogen phosphate monohydrate (NaH$_2$PO$_4$·H$_2$O), sodium nitrate (NaNO$_3$) and human serum

albumin (HSA) were purchased from MilliporeSigma (Darmstadt, Germany). The carrier liquid for AF4 and solution for sample preparation was prepared with water purified through a Milli-Q Plus purification system (Millipore Co. Ltd., Billerica, USA, resistance = 18.2 MΩ/cm). Model protein (henceforth referred to as GA-Z) was provided by Swedish Orphan Biovitrum AB (publ.) (Stockholm, Sweden). The GA-Z was designed to have one HSA binding site and one target molecule binding site with 108 amino acids (MW = 11.5 kDa) and it was produced in *E. Coli.*

## Asymmetrical flow field-flow fractionation (AF4)

The asymmetrical flow field-flow fractionation (AF4) used in this work was an Eclipse 3+ system (Wyatt Technology, Dernbach Germany) coupled online with a UV detector (UV-975, Jasco Corporation, Japan) set at 280 nm, a multi-angle light scattering (MALS) detector (DAWN HELEOS II, Wyatt Technology), and a differential refractive index (dRI) detector (Optilab T-rEX, Wyatt Technology). The AF4 channel was trapezoidal with a tip-to-tip length of 26.5 cm and the inlet and outlet widths of 2.2 and 0.6 cm respectively and was equipped with a 350 μm thick Mylar spacer and a regenerated cellulose (RC) membrane (molecular weight cut-off of 10 kDa, Millipore, Bedford, USA). The AF4 carrier liquid was 25 mM phosphate buffer with/without 125 mM NaCl at pH 7.0 and was pumped into the AF4 channel using an Agilent 1200 HPLC pump equipped with an auto-sampler and an inline vacuum degasser (Agilent Technologies, Waldbronn, Germany). The channel flow rate was kept constant at 1.0 mL/min, while the cross-flow rate was kept constant at 4.5 mL/min for 30 min. The injected volumes were 150 μl for the GA-Z HSA mixtures and 75 μl for HSA and GA-Z. The concentrations were 2 mg/ml for GA-Z and HSA samples, and for the mixtures 2 mg/ml HSA was mixed 1:1 with GA-Z solutions with concentrations of 0.01, 0.05, 0.1, 0.3, 0.5, 1.0, and 2.0 mg/mL, giving molar ratios between GA-Z and HSA of 0.03, 0.142, 0.28, 0.85, 1.42, .83 and 5.66. The channel was washed with the carrier liquid for 10 min without cross-flow at the end of each run. All AF4 experiments were performed at room temperature. The collection and processing of AF4 data were performed using the ASTRA software (ver. 6.1.17, Wyatt Technology) with $d_n/d_c$ for HSA and GA-Z of 0.185 mL/g and 0.196 mL/g. The $d_n/d_c$ value of GA-Z were determined by measuring concentrations in triplicates for the range between 0.0625 mg/ml and 2 mg/ml, using a batch mode dRI detector and then determining the slope with linear regression. In all cases, the Berry method was used to fit the light scattering data [24, 25]. The Berry fit used for molecular weight determination is calculated based on the LS signals measured (by multi-angle light scattering) in the ASTRA software (Wyatt) which calculates uncertainties for all reported quantities. By analyzing the baseline data at the beginning and end of the fractogram (or chromatogram), ASTRA determines the statistical fluctuation in each detector's output, including all photodiodes signal from MALS and the AUX signal from UV or dRI detectors. The AUX signal used for the error of LS fitting determination is the signal of the concentration detector used the calculation of molecular weight. In this work, we used the dRI detector to determine the molecular weight of the sample. Therefore, the error calculation of LS fitting used the AUX signal from the dRI detector. Each detector is weighted based on the fluctuations (noise) seen in the first and last 10% of the data points, up to 100 data points. Whichever end is least noisy is used to calculate the weighting factor. The error bars in the analysis plot do not represent this weighting factor directly. The analysis plot involves performing an nth order polynomial fit to $R_\theta / K^* c$ (for the Conventional Method), $K^* c / R_\theta$ (for the Zimm (Reciprocal) Method) [26], (for the Berry (Square Root) Method) [24]. The error bar calculation therefore involves the weighting factor, the normalized $R_\theta$ value as well as a

**Table 1.**

| Sample | c, (mg/ml) | $R_h$ (nm) from AF4 theory | Apparent MW (kDa) from AF4-MALS |
|---|---|---|---|
| GA-Z (whole angle) | 2 | 2.8 ± 0.01 | 13.2 ± 0.11 |
| GA-Z (low angle) | 2 | 2.8 ± 0.01 | 22.6 ± 1.49 |
| GA-Z (high angle) | 2 | 2.8 ± 0.01 | 12.0 ± 0.01 |
| HSA monomer | 2 | 3.3 ± 0.02 | 66.3 ± 0.22 |
| HSA dimer | 2 | 4.8 ± 0.03 | 134.6 ± 2.54 |
| free GA-Z (for GA-Z, with high angle) | Molar ratio 0.03 | n.d[1] | n.d[2] |
|  | Molar ratio 0.14 | n.d[1] | n.d[2] |
|  | Molar ratio 0.28 | n.d[1] | n.d[2] |
|  | Molar ratio 0.85 | n.d[1] | n.d[2] |
|  | Molar ratio 1.42 | 2.7 ± 0.01 | 11.3 ± 0.15 |
|  | Molar ratio 2.83 | 2.8 ± 0.02 | 11.1 ± 0.25 |
|  | Molar ratio 5.66 | 2.8 ± 0.01 | 12.3 ± 0.05 |
| GA-Z-HSA (for monomer HSA) | Molar ratio 0.03 | 3.4 ± 0.01 | 66.1 ± 0.75 |
|  | Molar ratio 0.14 | 3.4 ± 0.02 | 66.7 ± 0.36 |
|  | Molar ratio 0.28 | 3.5 ± 0.01 | 67.2 ± 0.21 |
|  | Molar ratio 0.85 | 4.0 ± 0.04 | 68.3 ± 0.23 |
|  | Molar ratio 1.42 | 3.9 ± 0.01 | 70.3 ± 0.33 |
|  | Molar ratio 2.83 | 4.0 ± 0.03 | 70.8 ± 1.25 |
|  | Molar ratio 5.66 | 4.1 ± 0.01 | 71.0 ± 0.24 |
| GA-Z-HSA (for dimer HSA) | Molar ratio 0.03 | 4.8 ± 0.01 | 130.4 ± 0.62 |
|  | Molar ratio 0.14 | 4.8 ± 0.02 | 130.3 ± 0.57 |
|  | Molar ratio 0.28 | 4.9 ± 0.01 | 133.2 ± 1.79 |
|  | Molar ratio 0.85 | 5.4 ± 0.01 | 137.4 ± 0.97 |
|  | Molar ratio 1.42 | 5.4 ± 0.01 | 144.8 ± 0.85 |
|  | Molar ratio 2.83 | 5.5 ± 0.02 | 144.5 ± 5.55 |
|  | Molar ratio 5.66 | 5.5 ± 0.01 | 153.2 ± 1.01 |

Dimensions of the proteins and complexes calculated from AF4 theory using the FFFHydRad 2.0 software. Apparent MW from AF4 measurements based on analyses of the signal MALS- detector.

* n.d[1] = no detection, n.d[2] = not determined (low light scattering signal).

concentration uncertainty factor and the $\chi^2$ value (different from the $\chi^2$ value for SAXS data fit above) returned from the fit. The different errors combine according to the usual rules for propagation of errors to yield a standard deviation (depending on calculation method) for each slice. These in turn allow calculation of uncertainties in the molar mass and size for each slice, and hence uncertainties in the calculated molar mass and size averages. These uncertainties are statistical only, and do not include any of the many possible systematic errors that may be present. All AF4 experiments were repeated three times for reproducibility and error calculation, and the error of AF4 results were less than 1% for all samples. Details of the errors for size and molecular weight estimations from AF4 are included in Table 1. From AF4 retention time ($t_r$) the hydrodynamic diameter ($D_h$) of a sample was calculated from AF4 theory using the FFFHydRad 2.0 software [27].

The fractograms were deconvoluted by fitting the data to a gaussian equation of first or second order using MatLab. The area under the peaks was calculated using numeric integration and the amount of GA-Z in each peak was calculated based on the assumption that the binding of GA-Z does not affect the monomer-dimer ratio of HSA in the samples. The bound fractions

were calculated according to Eqs 1–3

$$GA - Z \; per \; HSA_{dimer} = ((AUC(mixture) - AUC(HSA_{dimer}) * Konv_{GA-Z})/C(HSA_{dimer}) \quad (1)$$

$$Free \; GA - Z \; per \; HSA_{tot} = (AUC(mixture) - AUC(HSA_{dimer}) * Konv_{GA-Z}/C(HSA_{tot}) \quad (2)$$

$$GA - Z \; per \; HSA_{monomer}$$
$$= (C(GA - Z \; added) - C(GA - Z \; free) - C(GA - Z \; in \; dimer))/C(HSA_{monomer}) \quad (3)$$

AUC = Area under curve for deconvoluted peaks, $Konv_{GA-Z}$ = conversion factor from area to concentration, C(M) = concentration of protein (M).

## Small-angle X-ray scattering (SAXS)

SAXS data were collected at the SOLEIL synchrotron in France as well as the Petra III synchrotron in Germany, at the highly automated beamlines SWING and P12 [28, 29]. The data were reduced using PyFAI [30] and subsequently normalized to the transmitted beam, set to absolute scale and the background scattering was subtracted. Primus [31] was used to average frames as well as estimate forward scattering $I(0)$, radius of gyration $R_g$, and pair-distance distribution functions $P(R)$. Primus also automatically calculates the excluded volume of the hydrated particle (Porod volume $V_p$), which when divided by 1.6 gives an estimate of the scattering protein's molecular weight [32]. Further data fitting and modeling were performed using Crysol [33], Coral [34] and EOM [35] with standard settings. Crysol can be used to directly fit experimental data with high or low-resolution three-dimensional structures such as those produced with X-ray crystallography and single particle electron microscopy reconstructions. Coral and EOM fits experimental data using high-resolution rigid bodies and low-resolution flexible regions. Each amino acid in a low-resolution flexible region is represented by a single sphere. While Coral attempts to minimize the discrepancy between experimental data and the theoretical scattering of a single model, EOM instead minimizes the discrepancy between experimental data and the average theoretical intensities of an ensemble of models varying by the conformation adopted by defined flexible regions. For a more detailed description of these methods we refer to their respective publications [34, 35]. For HSA we used subunit A in PDB ID: 1AO6 as a model for fitting our HSA SAXS data. For the complex between GA-Z and HSA we used PDB ID: 1TF0 as a template for modeling the SAXS data, and we kept the GA-domain and HSA dimer interface in the crystal structure intact. For GA-Z we divided the structure into two rigid bodies, namely the GA-domain (residues 20–65 of PDB ID: 1GJS) and the Z-domain (residues 1–55 of model 1 in PDB ID: 1Q2N) which were linked by a flexible 7 amino acid residues long linker when using Coral and EOM. For modeling GA-Z dimers we applied P2 symmetry. All data sets were also modeled with DAMMIF [36], where 20 unique *ab initio* models were made for each data set. DAMMIF generates low resolution models of proteins using densely packed beads. It attempts to minimize the discrepancy between experimental data and theoretical intensities for the *ab initio* model. The *ab initio* model's excluded volume $V_a$ was divided by 2 to estimate the molecular weight [32]. DAMAVER [37] was used to make an averaged model of the 20 unique DAMMIF models. In all methods minimizing the discrepancy between experimental data and theoretical intensities was performed by minimizing $\chi$ or $\chi^2$:

$$\chi = \sqrt{\frac{1}{N-1} \sum_j \left[ \frac{(I_{exp}(S_j) - cI_{calc}(S_j))}{\sigma(S_j)} \right]^2}$$

Where N is the number of experimental points, c is a scaling factor, and $I_{exp}(s)$, $I_{calc}(s)$, $\sigma(S_j)$ are the experimental intensity, theoretical intensity, and experimental error at the momentum transfer $S_j$, respectively.

To investigate concentration-dependent effects on the structure GA-Z was measured at 3, 6, and 9 mg/mL. All samples were in 25 mM sodium phosphate buffer pH 7, 125 mM NaCl, and were centrifuged for 10 min at 10k RPM before loading the sample plate used by the beamline auto-sampler, which introduce the samples or their corresponding buffer to the SAXS capillary. Between samples, the SAXS capillary is automatically cleaned with water and air-dried. GA-Z (at 10 mg/mL), HSA (at 10 mg/mL), and the GA-Z-HSA mixture (at 5 mg/mL and 20 mg/mL, respectively) were also applied in separate injections on an Agilent Bio SEC-3 300Å size-exclusion chromatography column connected to an Agilent HPLC-system which was in turn connected directly to the SAXS capillary. The mobile phase used was 25 mM sodium phosphate buffer pH 7, 125 mM NaCl. SAXS data of the eluate were continuously collected as it flowed through the SAXS capillary. For subtraction, we used the average signal from the buffer coming off the column before any peaks appeared in the chromatogram. To select which buffer frames to average we used the data comparison algorithm in Primus [31].

To calculate a shape factor for the samples, we divided the radius of gyration determined from SAXS by the hydrodynamic radius determined from AF4-MALS. For a compact sphere the shape factor would be 0.77, as described by Burchard *et al.* [38]. For GA-Z-HSA samples the $R_h$ value from AF4-MALS at molar ratio 1.42 was used as it corresponds to the molar ratio used in SEC-SAXS experiments.

## Results

### Asymmetrical flow field-flow fractionation (AF4)

Fig 1a and 1b shows the AF4-UV-MALS-dRI fractograms and molecular weight (MW) of HSA and GA-Z obtained in 25 mM phosphate buffer at pH 7. As shown in HSA fractograms (Fig 1a) the monomer and dimer of HSA elute from 3.8 to 5.6 min and 5.8 to 7.2 min, respectively. The peaks of the chromatograms are at 4.5 and 6.5 min which corresponds to a hydrodynamic radius ($R_h$) of 3.2 (monomers) and 4.7 nm (dimers) calculated from peak maxima elution time by AF4 theory. See Table 1 for more details on sizes derived for HSA from AF4. Using MALS, the MW for the monomer and the dimer of HSA were estimated to be 66.2 kDa

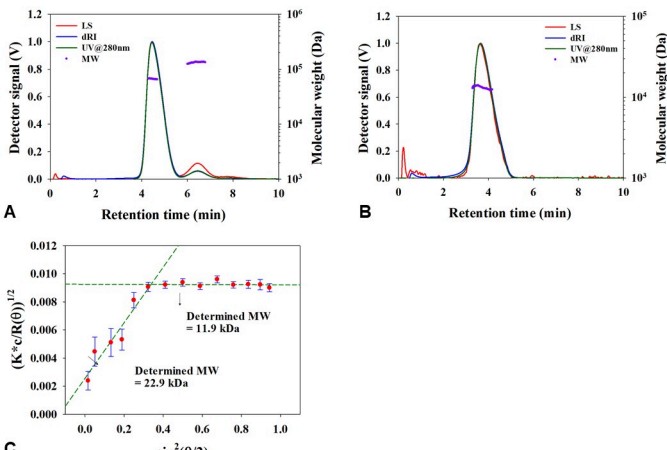

**Fig 1. AF4 fractograms and molecular weight (MW) of (a) HSA, (b), GA-Z and (c) Berry plot of GA-Z in 25 mM phosphate buffer at pH 7.**

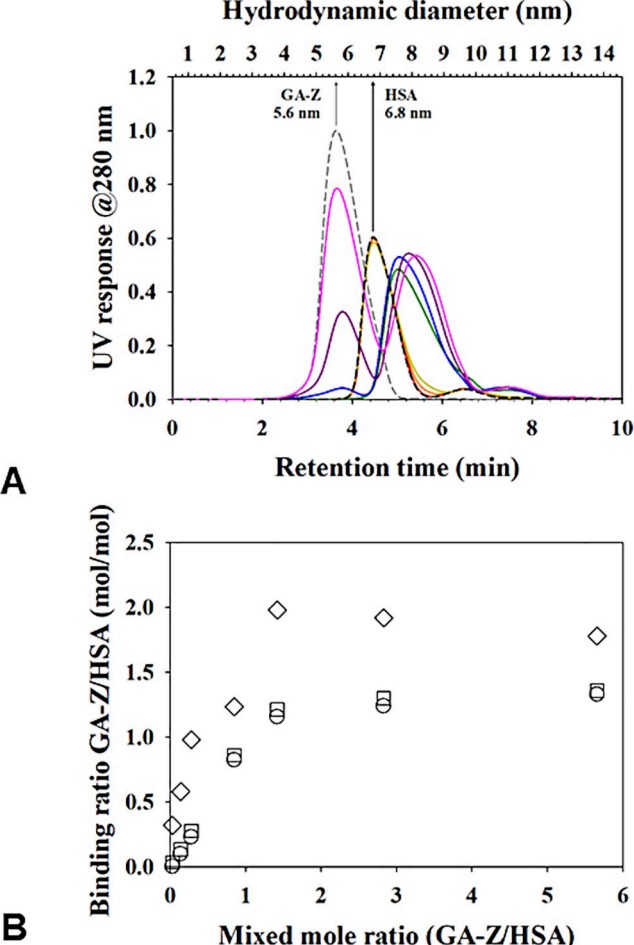

**Fig 2. (a)** AF4-UV fractograms and hydrodynamic radius (dH) of the complex between GA-Z and HSA with different mixed mole ratios The gray and black dashed lines are fractograms of pure GA-Z and HSA, and the solid lines are fractograms of each mixed mole ratio (red = 0.03, orange = 0.14, dark yellow = 0.28, green = 0.85, blue = 1.42, purple = 2.83, and pink = 5.66). **(b)** The calculated binding mole ratio from each UV detector response GA-Z bound to total amount of HSA (square), GA-Z bound to monomeric albumin (circle) GA-Z bound to dimeric HSA (diamonds).

and 138 kDa respectively. Additionally, to confirm the content of monomers and dimers of HSA a UV detector response at 280 nm was monitored. The contents of HSA monomer and dimer under the measurement conditions were determined to 93% and 7% respectively.

Fig 1b shows that GA-Z is eluted from 3 to 5 min, with a peak elution time corresponding to $R_h$ = 2.6 nm. See Table 1 for more details on sizes derived for GA-Z from AF4. However, molecular weight does not tend to increase with increasing retention time due to the co-elution of the monomer and dimer of GA-Z. In addition, the evidence of co-elution can be found by the 1st degree Berry fitting plot at peak maxima of GA-Z (from the light scattering signals of 14 angles). The MW was determined to be 11.9 kDa from the higher angle fit (8 points, 90° to 147°) and 23.9 kDa from the low angle fit (6 points, 32° to 81°) which are similar to nominal MW of monomer and dimer GA-Z (nominal MW of GA-Z monomer = 11.5 kDa, and dimer = 23.0 kDa), and provide clear evidence of co-elution in AF4 separation.

Table 1 gives details on sizes derived for HSA-GA-Z complex from AF4. Fig 2 shows the AF4 fractograms and binding mole ratio between GA-Z and HSA from different mole ratios (GA-Z/HSA = 0.03, 0.14, 0.28, 0.85, 1.42, 2.83 and 5.66). For comparison, pure HSA at the

same concentration as in the mixtures and GA-Z at the highest concentration used in the experiments are also shown.

At the lower GA-Z/HSA mole ratios (0.03–0.85) no free GA-Z could be detected (Fig 2a), which otherwise elutes between 3 to 5 min retention time. Instead, we found two peaks around the retention times of HSA (monomer and dimer), but with small tails appearing at GA-Z/HSA mole ratio of 0.14. The tails can be seen as a slight increase in maximum hydrodynamic diameter ($D_{h, max}$) for the peaks, see Table 1. We interpret this as low amount of GA-Z bound to both the monomer and dimer of HSA. The change is small due to the low amount of GA-Z present relative to HSA. For the dimer, the change is barely noticeable, but a trend can be seen when overlaying the graphs for pure HSA and the mixtures. At a GA-Z/HSA mole ratio of 0.85, the retention time of monomeric and dimeric HSA shifted from 4.4 to 5 min and 6.5 to 7.5 min, respectively. At GA-Z/HSA mole ratio $\geq$ 0.85 the HSA peak has shifted further from 5 to 5.5 min retention time and a growing peak of GA-Z also appears, indicating that there are free GA-Z in the samples. Table 1 presents the $R_h$ and $D_{h, max}$ for the peaks seen for the mixtures of GA-Z and HSA. The binding mole ratio was calculated from the fractogram in Fig 2a, and the binding isotherm is shown in Fig 2b.

## Small angle X-ray scattering of GA-Z

As is standard in SAXS experiments, we first investigated whether GA-Z showed concentration-dependent behavior. Thus, we measured GA-Z at three different concentrations: 3, 6 and 9 mg/mL (Fig 3). As shown in Table 2, the lowest concentration of GA-Z had an average radius of gyration ($R_g$) of 2.8 nm, while the two higher concentrations had an average $R_g$ of 3.2 nm. The estimated molecular weight increased only slightly with concentration and corresponded to that of a dimer at all concentrations. We could thus barely measure concentration-dependent behavior between 3 and 9 mg/ml of GA-Z.

## GA-Z SEC-SAXS

From previous experiments (unpublished data), and also in this work using AF4, we have found that GA-Z appears as both monomer and dimer in solution. To be able to model the GA-Z solution structure we needed to separate monomers from dimers. To this end, we used an HPLC size-exclusion column, directly linked to the SAXS sample cell. GA-Z eluted from the column as a single peak with a small tail (Fig 4). We measured SAXS continuously across the peak and the tail and in data analysis found that the peak (Fig 4) was GA-Z dimers. Using the data comparison algorithm in Primus [27] we could identify SAXS data frames within the elution peak similar to each other for averaging and improvement of the signal-to-noise ratio. The SAXS data result of the GA-Z SEC-peak (Fig 5a) was a scattering particle with an $R_g$ of 2.86 nm, a $D_{max}$ of 12.0 nm, a Porod volume of 34 nm$^3$, and a molecular weight based on the Porod volume of 22 kDa, which is compatible with a GA-Z dimer (Table 2). This averaged dataset from SEC we now call GA-Z dimer dataset. We interpret the tail of the main peak to arise from GA-Z monomers. However, a Guinier plot of the SAXS data revealed non-linear behavior strongly indicative that the protein in this fraction had started to aggregate (data not shown).

*Ab initio* modeling (using P2 symmetry) of the GA-Z dimer eluting from the size exclusion chromatography experiment resulted in an elongated shape with an excluded volume of 46 nm$^3$ (Fig 5b), which is equal to an estimated molecular weight of 23 kDa. Interestingly, each of the 20 individual models produced by DAMMIF had similar core structure to that of the average model produced by DAMAVER, but many of them were otherwise different to each other in that their arms occupied a large conformational space (S1 Fig). We continued modeling of

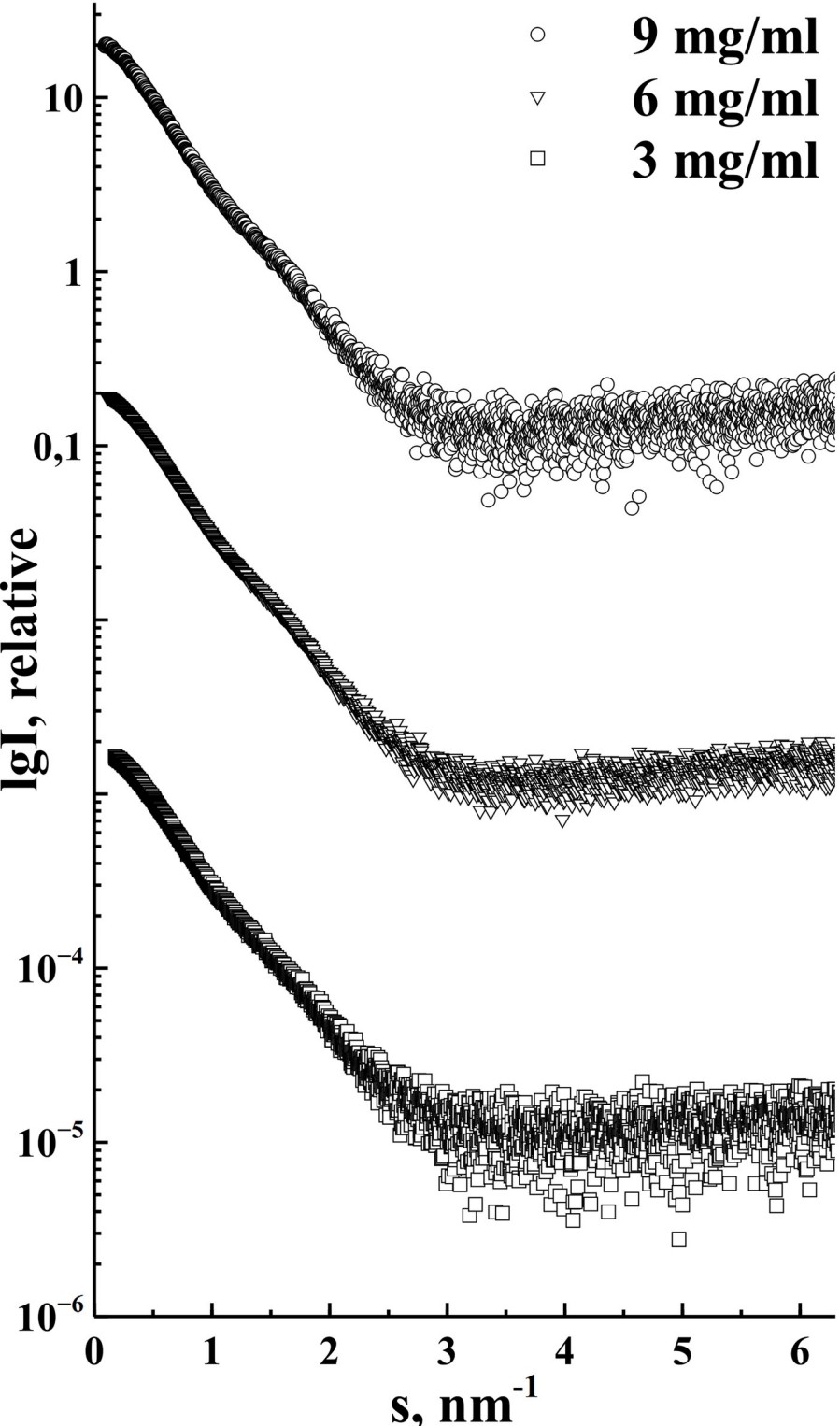

**Fig 3. Experimental SAXS data of GA-Z at three different concentrations 9 mg/ml (circles), 6 mg/ml (triangles), 3 mg/ml (squares).**

**Table 2.**

| Sample | c, (mg/ml) | $R_g$, (nm) | $D_{max}$, (nm) | $V_p$, (nm³) | $MW_p$, (kDa) | $MW_s$, (kDa) | Shape factor ($R_g/R_h$) |
|---|---|---|---|---|---|---|---|
| GA-Z | 3 | 2.84±0.07 | 11.6±1.4 | 38 | 24 | 25 | 1.0 |
| GA-Z | 6 | 3.15±0.11 | 14.2±0.8 | 40 | 25 | 28 | 1.1 |
| GA-Z | 9 | 3.15±0.11 | 14.9±0.9 | 44 | 28 | 29 | 1.1 |
| GA-Z | SEC | 2.86±0.68 | 12.0±0.2 | 34 | 22 | - | 1.0 |
| HSA | SEC | 2.81±0.01 | 8.7±0.3 | 100 | 63 | - | 0.85 |
| HSA-dimer | SEC | 4.08±0.05 | 12.4±0.5 | 213 | 133 | - | 0.85 |
| GA-Z-HSA 1 | SEC-1 | 3.18±0.04 | 12±0.3 | 125 | 78 | - | 0.82 |
| GA-Z-HSA 2 | SEC-2 | 3.23±0.04 | 12±0.5 | 139 | 87 | - | 0.83 |

Parameters for GA-Z and HSA samples as calculated from SAXS data. Concentration (c), pH, radius of gyration ($R_g$), maximum size ($D_{max}$), Porod volume ($V_p$), and molecular weight estimations based on Porod volume ($MW_p$) and BSA standard measurement ($MW_s$). The shape factor is the ratio between radius of gyration and hydrodynamic radius as estimated from AF4-MALS.

the GA-Z solution structure using the Coral software [32] and were able to model a P2 symmetrical dimer (Fig 5b) of GA-Z that fit the experimental data (Fig 5a) with a $\chi^2 = 1.0$. The model from Coral resulted in the dimeric interface to be between two Z-domains.

Even though Coral found a very good model to fit the GA-Z dimer data, *ab initio* modeling suggested GA-Z needed to be described as a flexible particle rather than a simple rigid body.

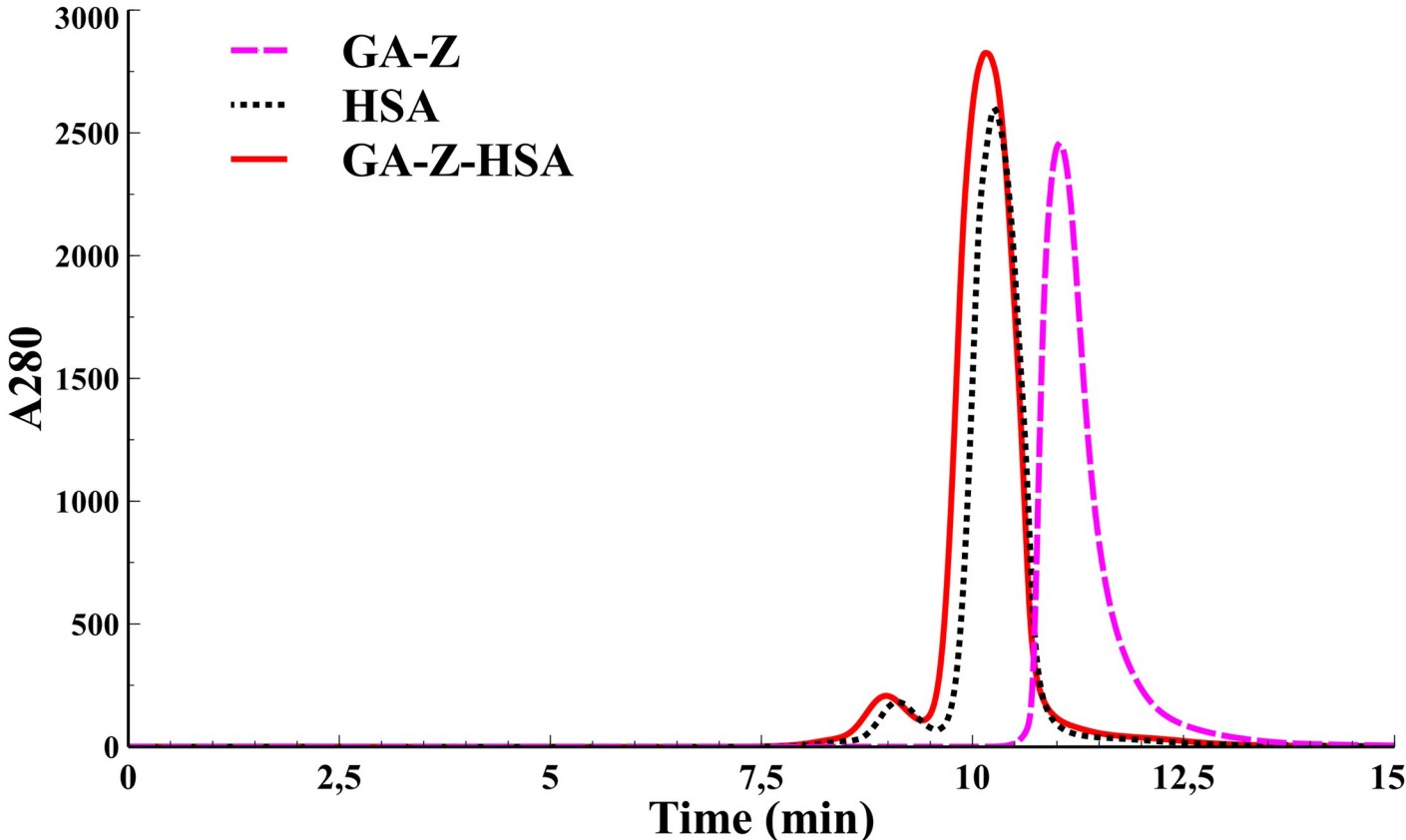

**Fig 4. Size-exclusion chromatography chromatogram of GA-Z (dashed magenta), HSA (dotted black), GA-Z-HSA (line red).**

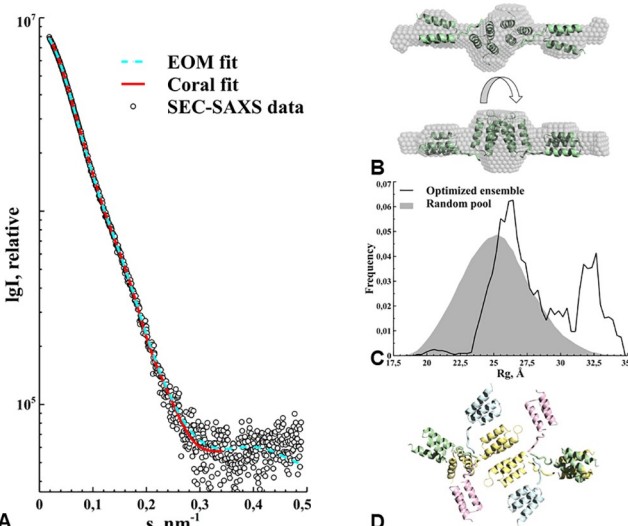

**Fig 5. SEC-SAXS data and modeling of the GA-Z dimer. (a)** Experimental data is shown as white circles with the fits using Coral (red line) and EOM (dashed blue line) modeling. **(b)** Coral rigid body model results corresponding to the fit in (a), superimposed on the ab initio model using DAMMIF on the same experimental data. The model is shown also after a 90 ° rotation. **(c)** Results from the EOM modeling, shown in grey filling is the random pool from which models could be chosen to fit the data in (a). The black line corresponds to the frequency of models with a certain radius of gyration that was used in the final fit in (a) to the experimental data. **(d)** Four representative models from the EOM modeling procedure, shown in ribbon representation.

Thus, we proceeded to model the GA-Z dimer using EOM. As with Coral the flexible region was defined as the seven amino acid long linker between the GA-domain and the Z-domain. Also, we decided to keep the dimer interface between the Z-domains generated by Coral as this appears to fit well with the average scattering volume calculated by *ab initio* modeling using DAMMIF and DAMAVER (Fig 5b). In this way EOM generated a pool of 10000 GA-Z dimers with P2 symmetry applied. The results showed a fit to the experimental data (Fig 5a) with a $\chi^2 = 0.86$. In the $R_g$ distribution (Fig 5c) at least two clear groups of conformations, one at the intermediate size and one at the extreme maximum size, can be seen in the pool of conformations. Upon closer inspection of the representative models from EOM (Fig 5d), we found that the linker between the GA-domain and the Z-domain is flexible enough to allow anything from longer elongated states to bend in a way that the two domains' helix bundles almost meet within a subunit.

### Small angle X-ray scattering of HSA and GA-Z-HSA

In preparation to study the isolated complex between GA-Z and HSA using SAXS, we first studied HSA alone as it eluted from the size exclusion column. We averaged the frames from the main peak (Fig 4) in the same way as with GA-Z, using data comparison in Primus. Using Crysol we could fit the HSA crystal structure from PDB ID 1AO6 (subunit A) to the averaged HSA SEC-SAXS experimental data (Fig 6a) with a $\chi^2 = 2.17$. The minor peak in the elution profile (Fig 4) had an estimated molecular weight of an HSA dimer, further parameters for HSA can be found in Table 2. This shows that our HSA was well prepared and in a native state, and we could continue with the GA-Z-HSA complex experiments using the crystal structure with PDB ID 1TF0.

The elution profile for GA-Z-HSA was similar to the HSA elution profile (Fig 4), but with a small shift in the peaks much like the AF4 experiments. This again shows that GA-Z bind to

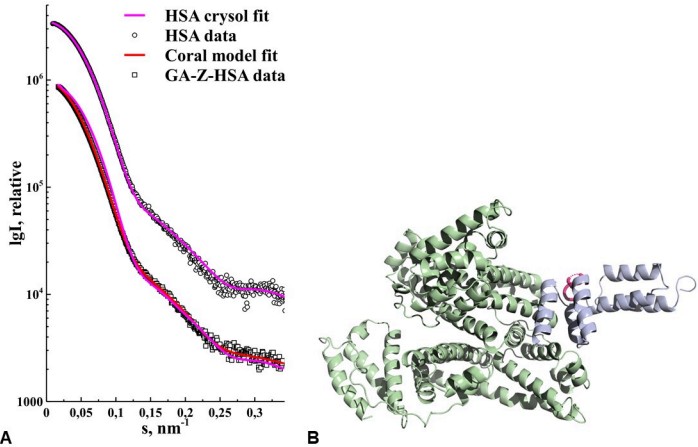

**Fig 6.** **(a)** HSA and GA-Z-HSA experimental data is shown as white circles and white squares, respectively. The corresponding fit to the HSA experimental data using Crysol and an X-ray crystallography model with PDB ID: 1TF0 is shown with a magenta line. For fitting to the GA-Z-HSA experimental we show both the Crysol fit (magenta line) using only HSA PDB ID: 1TF0 as well as that from the Coral rigid body model (red line), for comparison. **(b)** The resulting rigid body model from Coral is shown in a ribbon representation with HSA in green and GA-Z in blue, the linker between the GA and Z-domains is shown in magenta. GA-Z is bound to the HSA DII domain.

HSA and stayed bound during the chromatography experiment. Using the same procedure as before, we averaged the frames in the main peak (Fig 4) and found two different sets of averaged data that will be referred to as GA-Z-HSA-1 and GA-Z-HSA-2 (Fig 6a). GA-Z-HSA-1 has an $R_g$ of 3.2 nm, a $D_{max}$ of 12 nm, a Porod volume of 125 nm$^3$, and an estimated molecular weight of 78 kDa based on the Porod volume (Table 2). The size of GA-Z-HSA-1 is thus equivalent to a GA-Z monomer bound to HSA. GA-Z-HSA-2 has an $R_g$ of 3.2, a $D_{max}$ of 12, a Porod volume of 139 nm$^3$, and an estimated molecular weight of 87 kDa based on the Porod volume. The size of GA-Z-HSA-2 is equivalent to a GA-Z dimer bound to HSA. As with HSA, the minor peak in the GA-Z-HSA elution profile (Fig 4) involves HSA dimers but now bound to GA-Z in different ratios. We did not analyze the minor peak further due to low signal-to-noise in the data set, arising from a heterogeneous SEC peak likely containing various combinations of HSA dimer bound to monomers and dimers of GA-Z.

*Ab initio* modeling of GA-Z-HSA-1 showed a spherical shape with an additional feature on the side. The core structure fit that of HSA, and we interpret the additional feature as bound GA-Z. Further modeling of GA-Z-HSA-1 involved using Coral software and three rigid bodies, HSA and the two domains in the GA-Z monomer bound together with a seven amino acid residues long linker. Searching the protein data bank (PDB) we found a crystal structure of HSA bound to a GA-domain (PDB ID: 1TF0). We decided to keep the GA-domain HSA interface from PDB ID: 1TF0 intact, and thus only modeled the placement of the Z-domain (PDB ID: 1Q2N). The resulting model (Fig 6b) fit the experimental data with a $\chi^2 = 1.12$ (Fig 6a).

*Ab initio* modeling of GA-Z-HSA-2 showed similar results as that of GA-Z-HSA-1 but with a slightly larger side feature, likely arising from bound GA-Z dimer (results not shown).

## Discussion

In this work, we have attempted to increase the knowledge as a foundation for further development of albumin-binding therapeutically active fusion proteins for prolonged circulation in blood. We used a model protein, GA-Z, with an albumin-binding domain to directly study the interaction with HSA in solution. From AF4 we could estimate the binding ratio of GA-Z to

HSA as well as determine the hydrodynamic radii and molecular weights of HSA, GA-Z, and the average complex at different ratios. From SAXS we get a more detailed structure of both the individual molecules and the complexes between different species. By comparing the radius of gyration with the hydrodynamic radius we get a shape factor that is based on both methods, and that complement the more detailed structure from the SAXS measurements.

HSA is a well-documented protein and therefore the results for HSA in this study can verify that the methodologies used are suited for the studies. It is well known that HSA at pH 7 contains both monomers and dimers [3], and both SAXS and AF4 gave a molecular weight estimation of HSA monomer and dimer that is in line with the literature (Table 2). Using AF4 it could also be determined that HSA monomers outweigh the number of HSA dimers by almost 9 to 1. The ratio between them can differ depending on the state of the HSA molecule [2], e.g. aged HSA preparations typically have more dimers. The value found here fits well with what has been found previously [2]. The SAXS measurements gave a radius of gyration of 2.8 nm for the monomer whereas AF4 gave a hydrodynamic radius of 3.2. This gives a $R_g/R_h$ ratio of 0.87 in agreement with the slightly oblate ellipsoid shape of HSA [39]. The SAXS size is slightly lower than what has been reported by for example Olivieril and Craievich [20]. It is well known that the presence of both monomers and dimers in a sample will affect the interpretation of SAXS data [19] and the difference found is probably due to that monomers and dimers were separated using on-line SEC in our experiments.

The authors have previously found that GA-Z forms dimers under the ambient conditions used in this study. However, due to co-elution of GA-Z monomers and dimers both in the SEC and in the AF4, we could not make an estimation of ratios between monomer and dimers, although, the presence of both could be confirmed from AF4 MALS detection (Fig 1c). This is surprising as one would expect higher resolution between a monomer and dimer. Most likely this is due to a very rapid monomer-dimer exchange of the protein. In the case of very fast monomer-dimer exchange it has been shown that it is difficult to separate the individual components of such protein complex [40, 41] and they tend to elute as a single asymmetrical peak. Average molecular weight estimation of GA-Z SAXS data collected at three different concentrations (3–9 mg/ml) suggested that the dimer form was most common in the whole concentration range. The estimated average molecular weight in bulk solution determined by SAXS without SEC is similar to what was estimated for the main part of the peak containing GA-Z in the SEC-SAXS experiments (Table 2). Interestingly, the tail of the peak, which would correspond to monomeric protein, was found to be aggregated. This suggests that monomeric GA-Z could be less stable. The reason for this aggregation is currently not known, but as the aggregation clearly occurs after the SEC-separation it could be due to radiation damage. This phenomenon needs further investigation to gain more insight on stability of monomeric versus dimeric GA-Z.

SAXS combined with the small nature of GA-Z domains makes it difficult to resolve whether the interface of the GA-Z dimer was between GA-domains or Z-domains. In this context it is interesting to note that in all models predicted by both Coral (Fig 5b) and EOM (Fig 5d), the dimer interface was always between the Z-domains of two GA-Z monomers, leaving the GA-domains out in solution, free to bind to HSA. It should be noted that it is unlikely that we can resolve the true dimer interface using SAXS alone, due to the inherent low resolution of the technique combined with the similar size and shape of the GA-domain and the Z-domain. However, as we have found that GA-Z dimers bind to HSA it seems reasonable that the interaction site is between the Z-domains. In addition, as discussed below, the AF4 results also indicate that dimers as well as monomers of GA-Z bind to HSA, which further supports these models.

Dimeric GA-Z appears to exist primarily in two conformations, one fully extended and the other with an intermediate size with respect to the total conformational space of the molecule as suggested by the EOM modeling software (Fig 5c). Thus, the linker that fuses the GA-domain to the Z-domain allows a wide range of movement. Also, the distinct dual conformational space of the GA-Z dimer could suggest that within the dimer there is an equilibrium between the two main conformations of the free GA-domains and the interacting Z-domains within the dimer. This intra-dimeric equilibrium of conformations could be a distinct property of the dimer and could contribute to a more stable structure as compared to the aggregating monomer mentioned above. Similarly, in a study on a fusion protein, combining an albumin binding domain and an integrin binding fibronectin scaffold protein, also confirmed dimerization [42]. Furthermore, the authors found that a longer linker correlated with increased affinity to HSA, and that C-terminal placement of the albumin binding domain correlated with increased stability of the fusion protein. This further supports the argument that the intra-dimeric equilibrium of these molecules is important for their stability and function. Further investigation of the impact of oligomeric state on the stability of GA-Z is currently ongoing. The presence of the non-spherical structure is also supported by the $R_g/R_h$ ratio of GA-Z of approximately 1.1, which indicates that the structure deviates from what would be expected of the 0.77 for a compact sphere [38]. In fact, 1.1 is above the range for a compact oblate ellipsoid but below that for a prolate one [39]. In studying the unfolded state of proteins Choy *et al* identified an increase in the $R_g/R_h$ with increasing unfolding and thus increasing flexibility of the protein [43]. Hence, the high value seen here is in line with the high degree of flexibility found in the modeling of the SAXS data. It has been shown that an increase in the ratio between the long and short axes of a prolate structure increases the shape factor of peptides [44].

As expected, mixing GA-Z and HSA at varying molar ratios gave rise to a GA-Z-HSA complex. The tailing of the HSA peak in the AF4 fractogram (Fig 2a) is evidence of complex formation at GA-Z/HSA molar ratio < 0.85. At the GA-Z/HSA molar ratio of 0.85, the entire HSA peak was shifted while still no free GA-Z was detected, indicative of a strong complex formation. The binding ratio of GA-Z to both HSA monomer and dimer was estimated from the fractogram. Since the peaks are not baseline separated, this estimation should be interpreted with some caution. Still, there are some important conclusions that can be made based on these calculations. Firstly, there is no larger growth of what is most likely the dimer peak of HSA, neither is there any new peaks appearing at longer elution times (Fig 2). We see this as a strong indication that binding of GA-Z does not change the ratios between HSA monomers and dimers and, further, that the binding does not induce the formation of any other higher-order oligomers. Secondly, the amount of GA-Z that seems to be bound to the dimer form of HSA is at least as much or even more on a molar basis, than what is bound to the monomer form. This result indicates that the dimer formation of HSA does not hinder the binding of GA-Z. Thirdly, the overall binding ratio found in this experiment was 1.3 GA-Z molecule per molecule of HSA, which could suggest that there is more than one binding site on HSA to a single GA-Z. However, the presence of GA-Z dimers in solution is most likely the reason for the deviation in the calculated binding ratio. This could also be confirmed by the SEC-SAXS results, which showed that both dimer and monomer forms of GA-Z bind to HSA. It was not possible to determine which oligomeric form that was dominant in the complex due to technical limitations in SAXS and SEC. When we modeled the complex between GA-Z and HSA no alterations were made to the conformation of HSA. We cannot rule out that no conformational changes occurs in HSA upon interaction with GA-Z, however, given the limited resolution of SAXS we can say that no major changes that affect the overall shape of HSA took place. The shape factor for the complex was 0.82, which indicates that the complex is more spherical than HSA alone. In Fig 6b, the binding site of GA-Z seems to rather strengthen the oblate

structure. However, considering the domain flexibility of the GA-Z together with presence of GA-Z dimers in the complex, a more spherical average shape is not unreasonable. As with AF4, our SEC-SAXS data also showed that GA-Z bound to HSA dimers. However, due to a low concentration in this fraction leading to low signal-to-noise ratio, it was decided not to proceed further with modelling of that data.

## Conclusions

We have shown that GA-Z binding to HSA can be studied by a combination of AF4 and SAXS. The GA-Z protein exhibits a monomer/dimer equilibrium and the results show that the dimer is the dominant species at pH 7. There are strong indications that the GA-Z dimer interface is not in the GA-region. The determined structure and monomer-dimer ratios for HSA were found to correspond well with previous data for this protein, and thus validate the methods used. GA-Z was found to bind both monomers and dimers of HSA. It was clear that at least part of the GA-Z was bound in its dimeric form to both monomers and dimers of HSA.

## Supporting information

**S1 Fig. 19 of the 20 DAMMIF *ab initio* models computed from GA-Z SEC-SAXS data.** Each model has a unique colour and was superimposed automatically using DAMAVER. The filtered average model (damfilt.pdb) from DAMAVER software is shown in red spheres on top of the 19 individual models.
(PDF)

## Acknowledgments

We acknowledge Synchrotron SOLEIL, SWING beamline, and Petra III, P12 beamline, for providing assistance and access to synchrotron radiation beamlines.

## Author Contributions

**Conceptualization:** Marie Wahlgren, Vilhelm Ek, Ulla Elofsson, Jonas Fransson, Lars Nilsson, Ann Terry, Christopher A. G. Söderberg.

**Data curation:** Jaeyeong Choi, Marie Wahlgren, Lars Nilsson, Ann Terry, Christopher A. G. Söderberg.

**Formal analysis:** Jaeyeong Choi, Marie Wahlgren, Christopher A. G. Söderberg.

**Funding acquisition:** Marie Wahlgren, Ulla Elofsson.

**Investigation:** Jaeyeong Choi, Marie Wahlgren, Ann Terry, Christopher A. G. Söderberg.

**Methodology:** Jaeyeong Choi, Marie Wahlgren, Vilhelm Ek, Ulla Elofsson, Jonas Fransson, Lars Nilsson, Ann Terry, Christopher A. G. Söderberg.

**Project administration:** Marie Wahlgren, Vilhelm Ek, Ulla Elofsson, Jonas Fransson, Ann Terry.

**Resources:** Marie Wahlgren, Ulla Elofsson.

**Supervision:** Marie Wahlgren, Vilhelm Ek, Ulla Elofsson, Jonas Fransson, Lars Nilsson, Ann Terry, Christopher A. G. Söderberg.

**Validation:** Jaeyeong Choi, Marie Wahlgren, Ulla Elofsson, Lars Nilsson, Ann Terry.

**Visualization:** Jaeyeong Choi, Marie Wahlgren, Christopher A. G. Söderberg.

**Writing – original draft:** Jaeyeong Choi, Marie Wahlgren, Vilhelm Ek, Ulla Elofsson, Jonas Fransson, Lars Nilsson, Ann Terry, Christopher A. G. Söderberg.

**Writing – review & editing:** Jaeyeong Choi, Marie Wahlgren, Ulla Elofsson, Lars Nilsson, Ann Terry, Christopher A. G. Söderberg.

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
