## [Decision Letter · Decision Letter 0]

30 Apr 2020

PONE-D-20-09307

Characterization of binding between model protein GA-Z and human serum albumin using asymmetrical flow field-flow fractionation and small angle X-ray scattering

PLOS ONE

Dear Dr. Söderberg,

Thank you for submitting your manuscript to PLOS ONE. After careful consideration, we feel that it has merit but does not fully meet PLOS ONE’s publication criteria as it currently stands. Therefore, we invite you to submit a revised version of the manuscript that addresses the points raised during the review process.

Both reviewers were given positive review on your work. However they have raised few minor comments. In your revision, you need to carefully address them and rebuttal the corrections.

We would appreciate receiving your revised manuscript by Jun 14 2020 11:59PM. To enhance the reproducibility of your results, we recommend that if applicable you deposit your laboratory protocols in protocols.io, where a protocol can be assigned its own identifier (DOI) such that it can be cited independently in the future. For instructions see: http://journals.plos.org/plosone/s/submission-guidelines#loc-laboratory-protocols

We look forward to receiving your revised manuscript.

Kind regards,

Rajagopal Subramanyam

Academic Editor

PLOS ONE

Journal Requirements:

'The research in this study was performed with financial support from Vinnova - Swedish Governmental Agency for Innovation Systems and the Swedish Research Council within the NextBioForm Competence Centre.'

'This project was funded by a Vinnova grant awarded to UE and MW (grant number 2018-04730, www.vinnova.se). The funders had no role in study design, data collection and analysis, decision to publish, or preparation of the manuscript.'

'The authors have declared that no competing interests exist.'

We note that one or more of the authors are employed by a commercial company: Swedish Orphan Biovitrum AB (Sobi)

5. Please include captions for your Supporting Information files at the end of your manuscript, and update any in-text citations to match accordingly. Please see our Supporting Information guidelines for more information: http://journals.plos.org/plosone/s/supporting-information

Reviewers' comments:

Reviewer's Responses to Questions

**Comments to the Author**

1. Is the manuscript technically sound, and do the data support the conclusions?

Reviewer #1: Yes

Reviewer #2: Yes

2. Has the statistical analysis been performed appropriately and rigorously? 

Reviewer #1: Yes

Reviewer #2: Yes

3. Have the authors made all data underlying the findings in their manuscript fully available?

Reviewer #1: Yes

Reviewer #2: Yes

4. Is the manuscript presented in an intelligible fashion and written in standard English?

Reviewer #1: Yes

Reviewer #2: Yes

5. Review Comments to the Author

Reviewer #1: Characterization of binding between model 1 protein GA-Z and human serum albumin

using asymmetrical flow field-flow fractionation and small angle X-ray scattering:byChoi

Review report on above manuscript

This paper described the ineractions of GA-Z protein with human serum albumin. The owrks is done carefully and up-to-date references are provided and has useful information for the reader of PLOS and can be published after minor revisions

1. what are the binding constants of GA-Z protein with HSA

2. does protein interaction alter HSA conformation.

3. What are the binding sites of protein on HSA

Reviewer #2: The work presented by the author aims at characterizing the binding between model protein GA-Z and human serum albumin using asymmetrical flow field-flow fractionation and small angle X-ray scattering. The work was well designed and accurately conducted.

1. A more rigorous comparison of the present study with the literature in all respects would enhance the impact of the present study.

2. The abstract should contain the significance of the results obtained. It is therefore advised to add more on significance at the end of the abstract.

3. The manuscript lacks the information on the number of the replicate set of experiments and error analysis.

4. The font size of the text inside the graphs should be increased so that they become more legible. The quality of the figures is not suitable for publication.

5. In Table. 1 fourth column ‘apparent MW (kDa) from AF4-MALS’. The word apparent should be capitalized.

6. The presentation and interpretation of results executed should be more vivid.

In this context, I recommend this article for publishing only after the minor revision.

6. PLOS authors have the option to publish the peer review history of their article (what does this mean?). If published, this will include your full peer review and any attached files.

Reviewer #1: No

Reviewer #2: No

---

## [Author Response · Author response to Decision Letter 0]

5 Nov 2020

We thank the reviewers for their valuable comments to our manuscript.

Please find below our response to reviewers' comments on the manuscript "Characterization of binding between model protein GA-Z and human serum albumin using asymmetrical flow field-flow fractionation and small angle X-ray scattering".

Reviewer #1

1. “What are the binding constants of GA-Z protein with HSA"

It is not possible to determine the binding constant based on the data that we present here. However, we added examples on GA-HSA binding constants in the revised manuscript.

2. “Does protein interaction alter HSA conformation.”

From the SAXS data we cannot see any major alterations to the general shape of HSA, and we have clarified this in the revised manuscript.

3. “What are the binding sites of protein on HSA"

GA-Z binds to the DII domain of HSA, this information has been included in the revised manuscript.

Reviewer #2

1. A more rigorous comparison of the present study with the literature in all respects would enhance the impact of the present study.

The following references have been included to the introduction and discussion parts of the manuscript for comparison. Zorzi et al. (2019), Alexander et al. (2007), Gapizov et al. (2019)

2. The abstract should contain the significance of the results obtained. It is therefore advised to add more on significance at the end of the abstract

The following sentence has been added to the abstract to lift the significance of this manuscript: “Furthermore, GA-Z binds to HSA both as a monomer and a dimer, and this, it can be expected to stay bound also upon dilution following injection in the blood stream.”

3. The manuscript lacks the information on the number of the replicate set of experiments and error analysis.

This has been added to Material and Methods in the revised manuscript.

4. The font size of the text inside the graphs should be increased so that they become more legible. The quality of the figures is not suitable for publication.

We have revised the figures with small fonts to increase their readability. We have also incrased the DPI of low quality figures.

5. In Table. 1 fourth column ‘apparent MW (kDa) from AF4-MALS’. The word apparent should be capitalized.

This has been revised.

6. The presentation and interpretation of results executed should be more vivid.

In this context, I recommend this article for publishing only after the minor revision.

We have rephrased the discussion of the manuscript in an attempt to make it more “vivid”, although the content have not been altered.

Yours sincerely,

Christopher Söderberg

---

## [Editor Report · Decision Letter 1]

6 Nov 2020

Characterization of binding between model protein GA-Z and human serum albumin using asymmetrical flow field-flow fractionation and small angle X-ray scattering

PONE-D-20-09307R1

Dear Dr. Söderberg,

We’re pleased to inform you that your manuscript has been judged scientifically suitable for publication and will be formally accepted for publication once it meets all outstanding technical requirements.

Kind regards,

Rajagopal Subramanyam

Academic Editor

PLOS ONE
---

## [Editor Report · Acceptance letter]

13 Nov 2020

PONE-D-20-09307R1 

Characterization of binding between model protein GA-Z and human serum albumin using asymmetrical flow field-flow fractionation and small angle X-ray scattering 

Dear Dr. Söderberg:

I'm pleased to inform you that your manuscript has been deemed suitable for publication in PLOS ONE. Congratulations! Your manuscript is now with our production department. 

Kind regards, 

on behalf of

Prof. Rajagopal Subramanyam 

Academic Editor

PLOS ONE